

# Quantum approximate optimization for hard problems in linear algebra

**Ajinkya Borle[1\*], Vincent E. Elfving[2] and Samuel J. Lomonaco[1]**

**1** CSEE Department, University of Maryland, Baltimore County, Baltimore, USA
**2** Qu & Co BV, PO Box 75872, 1070 AW, Amsterdam, the Netherlands

\* aborle1@umbc.edu

## Abstract

The quantum approximate optimization algorithm (QAOA) by Farhi et al. is a quantum computational framework for solving quantum or classical optimization tasks. Here, we explore using QAOA for binary linear least squares (BLLS); a problem that can serve as a building block of several other hard problems in linear algebra, such as the non-negative binary matrix factorization (NBMF) and other variants of the non-negative matrix factorization (NMF) problem. Most of the previous efforts in quantum computing for solving these problems were done using the quantum annealing paradigm. For the scope of this work, our experiments were done on noiseless quantum simulators, a simulator including a device-realistic noise-model, and two IBM Q 5-qubit machines. We highlight the possibilities of using QAOA and QAOA-like variational algorithms for solving such problems, where trial solutions can be obtained directly as samples, rather than being amplitude-encoded in the quantum wavefunction. Our numerics show that even for a small number of steps, simulated annealing can outperform QAOA for BLLS at a QAOA depth of $p \leq 3$ for the probability of sampling the ground state. Finally, we point out some of the challenges involved in current-day experimental implementations of this technique on cloud-based quantum computers.

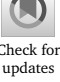
# 1 Introduction

The application of quantum computing to hard optimization problems is a candidate where quantum computing may eventually outperform classical computation [1–6]. At the time of writing this paper, noisy intermediate scale quantum (NISQ) computers [5] are being developed by several firms and research groups [7–13]. The two main approaches to quantum optimization are (i) the quantum annealing (QA) physical heuristic [6] and (ii) quantum approximate optimization algorithm (QAOA) [4] on the gate-model quantum computer [1].

In this paper, we explore and propose the use of QAOA for hard problems in linear algebra. In particular, we focus on the problem of binary linear least squares (BLLS) [14–16], a subroutine for solving certain types of non-negative matrix factorization (NMF) problems [17, 18], for example, the problem of non-negative binary matrix factorization (NBMF) [17, 19]. These are linear dimensionality reduction (LDR) tools useful in machine learning for learning a basis that would be used to model data [20], such as learning facial features in computer vision [21], topic modeling in documents [22] and properties of astronomical objects [23, 24]. Further-

more, BLLS and its variant (binary compressive sensing [25]) have applications in signal processing [16, 25], image processing [15, 26] and can be a building block for other problems in linear algebra [27–32]. We hope that our work provides insights to fellow researchers to further explore the use of NISQ era methods [4, 33] for problems in linear algebra and numerical computation. The outline of our results are as follows:

- The technique seems to scale well in optimizing the expectation value as the problem size $n$ increases (for a fixed QAOA circuit depth $p$). This corresponds to getting 'good solutions' fast but not necessarily the most optimal one (refer Section 3.4.2).

- The probability of finding the optimal solution appears to drop exponentially as the problem size increases (for a fixed QAOA circuit depth $p$). QAOA outperformed random sampling, but was beaten by simulated annealing. Further and extensive tests would be require to show if QAOA can deliver a quantum advantage for this criterion (refer Section 3.4.4).

- The implicit filtering (ImFil) classical optimizer backend [34] used for optimizing the QAOA parameters is robust to shot-noise (refer Sections 3.4.1, 3.4.3 and 3.4.6).

- Given a specific realistic quantum hardware noise model, a dense device coupling map would drastically reduce the errors in approximating the expectation value for this problem, when compared against a sparse one (refer Section 3.4.5).

- When compared against older quantum devices, the newer ones seem to be getting better at approximating the expectation value for a given set of QAOA parameters. However, the results they produce are still very noisy for QAOA to function effectively (refer Sections 3.4.1 and 3.4.7).

In Section 2, we cover the necessary background and related work for our paper. Section 3.1 is about formulating the BLLS problem into a classical Ising problem and Section 3.2 goes over the details of the gates required for the QAOA ansatz. The experiments, results and discussion are detailed in Sections 3.3, 3.4 and 3.5 respectively. We finally conclude our paper in Section 4. We also have Appendices to complement and support the information in the main paper.

## 2 Background and related work

### 2.1 Background

#### 2.1.1 The binary linear least squares (BLLS) problem

Given a matrix $A \in \mathbb{R}^{m \times n}$, an unknown column vector of variables $x \in \{0, 1\}^n$ and a column vector $b \in \mathbb{R}^m$ (Where $m > n$). The linear BLLS problem is to find the $x$ that would minimize $\|Ax - b\|$ the most. In other words, it can be described as:

$$\arg \min_x \|Ax - b\|. \tag{1}$$

The motivation behind choosing the BLLS problem as a target problem is twofold: firstly, it is an NP-Hard problem [16] that makes it a relevant target for potential speedup. Secondly, it can act as a building block for other hard problems in linear algebra, such as the Non-negative Binary Matrix Factorization [17]. One reason why one may view BLLS a building block for other problems is because multiple binary variables can be clubbed together for a fixed point

approximation of a real variable [18, 27–30, 35]. Amongst these, there are some problems that are NP-hard for which an approximate solution would be acceptable [18, 35]. In these cases, QAOA may be able to provide an improvement in finding approximate solutions against classical solvers and increase the probability of sampling the best solution.

### 2.1.2 Non-negative matrix factorization (NMF) and variants

NMF is a linear dimensionality reduction (LDR) tool that helps in data analysis. Given a matrix $V \in \mathbb{R}_{\geq 0}^{m \times n}$ of $n$ data points of dimension $m$, NMF extracts a basis $W \in \mathbb{R}_{\geq 0}^{m \times r}$ containing $r$ elements such that the linear space spanned by the basis $W$ approximates the datapoints in $V$ as best as they can [20]. In other words, for some matrix $H \in \mathbb{R}_{\geq 0}^{r \times n}$, $V \approx W \times H$. The restrictions to non-negative number makes the factored matrices sparse and easily interpretable for certain domains where non-negative feature vectors do not apply [20]. Few such examples are:

**Computer vision:** NMF is useful for facial recognition as the problem suffers from high dimensionality of data. Here, NMF can extract a set of features $W$ from the input/training data (matrix $V$). Then for an image unseen by the model before (test), represented by column vector $v^*$, we'll use $W$ and see if we can find some non-negative column vector $h^*$ such that $v^* \approx W \times h^*$. NMF is found to be more robust than another LDR technique, Principal Component Analysis or PCA (which doesn't restrict the data to be non-negative), for image occlusions [21].

**Astronomy:** NMF has applications in astronomy primarily due to astrophysical signals being non-negative. As an example, NMF was used on spectroscopic observations [23] and the direct imaging observations [24] in order to study the common properties of astronomical objects and post-process astronomical observations.

**Text mining/Topic modeling:** Here, text documents can be converted into non-negative column vectors. One way to do this is by considering a dictionary of $m$ words. Each matrix element $V_{ij}$ in the dataset $V$ would denote the number of occurrences of the $i$th word in the $j$th document. Applying NMF, a basis of topics $W$ can be generated along with the matrix $H$ that suggests the importance of each of the basis topics for each document (i.e. column) in $V$. This is useful to automate the extraction of abstract topics in a given set of documents [22].

The downside of NMF is that the factorization is NP-Hard in the general case [36].

**Non-negative binary matrix factorization (NBMF)** is a specialized version of the NMF problem; where entries in matrix $H$ are restricted as $H \in \{0,1\}^{r \times n}$. Here the matrix $H$ is even sparser than in the general NMF [17]. The features learnt by the matrix $W$ would be more holistic against those in NMF. As an example, In facial feature extraction, NMF is able to learn

---

**Algorithm 1** Alternating least squares for NBMF (pseudocode).

---

1: **procedure** MAIN($V$)                                      ▷ V is the matrix to be factorized
2:      Randomly initialize the matrix $H \in \{0,1\}^{r \times n}$
3:      **while** not converged **do**
4:          **for** row i from 1 to $m$ **do**
5:              $(W_i)^T \leftarrow \arg\min_{W_i^T} \|V_i^T - H^T (W_i)^T\|_2$ such that $W \in \mathbb{R}_{\geq 0}^{m \times r}$
6:          **end for**
7:          **for** column j from 1 to $n$ **do**
8:              $H_j \leftarrow \arg\min_{H_j} \|V_j - W H_j\|_2$ such that $H \in \{0,1\}^{r \times n}$
9:          **end for**
10:      **end while**
11:      **return** $W$,$H$
12: **end procedure**

---

parts of faces from a given dataset, whereas the features extracted by NBMF are closer to a complete face [17].

BLLS can be used in order to solve the NBMF variant by using the alternating least squares method [37].

In Algorithm 1, line 5 is solved classically since efficient algorithms exist for it [38], and would probably be applied with some kind of regularization since NMF and its variants tend to generally be ill-posed problems [17, 39]. It is line 8 that is solved using BLLS (where QAOA would be applied).

In the past, quantum annealing was used as a subroutine within this algorithm to solve NBMF and other NMF related problems [17, 18]. Based on our work with this paper, QAOA can be an alternative to quantum annealing for NBMF, which can be explored in the future.

### 2.1.3 Quadratic unconstrained binary optimization (QUBO)

The QUBO objective function is as follows,

$$F(q) = \sum_a v_a q_a + \sum_{a<b} w_{ab} q_a q_b, \tag{2}$$

where $q_a \in \{0, 1\}$, $v_a$ and $w_{ab}$ are real coefficients for the linear and quadratic parts of the function respectively. The QUBO objective function is NP-hard in nature [40]. One salient feature of this objective function is that many application domain problems map naturally to QUBO [17, 18, 27, 41, 42]. In the process of applying BLLS to gate model quantum devices, we use the QUBO formulation as an intermediate stage of expressing the problem.

### 2.1.4 The quantum approximate optimization algorithm (QAOA)

In 2014 Farhi et al. proposed an algorithm that uses both quantum and classical computation for solving optimization problems [4]. The potential advantage of using this algorithm is that it can be implemented by using low depth quantum circuits [43], making it suitable for NISQ devices. We here briefly summarize the QAOA formalism applied to binary optimization problems. For the required preliminaries of quantum computing, the authors recommend the textbook by Nielsen and Chuang [1].

One popular method of encoding an optimization problem to be solved using QAOA, is to first formulate the problem as an Ising Objective function.

$$F(\sigma) = \sum_a h_a \sigma_a + \sum_{a<b} J_{ab} \sigma_a \sigma_b, \tag{3}$$

$$\text{where } \sigma_a = 2q_a - 1. \tag{4}$$

Here $\sigma_a \in \{-1, 1\}$, $h$ and $J$ are coefficients associated with individual and coupled binary variables respectively. The Ising model is a popular statistical mechanics model, associated primarily with ferromagnetism [44]. Because it has been shown to be NP-Complete in nature [45], the objective function associated with it can be used to represent hard problems [46]. It is important to note here that we still don't know if quantum computing can help solve NP-Complete problems efficiently [47]. Our hope for quantum algorithms, at the very least, is to be able to compete with classical heuristics when it comes to certain classes of hard problems.

The problem then would be to maximize or minimize Eqn(3), depending on how it is set up. The (classical) Ising Hamiltonian, which naturally maps the Ising objective Eqn(3) to

**Algorithm 2** Quantum approximate optimization algorithm for minimization (pseudocode).

```
 1: procedure MAIN(B̂, Ĉ, p)                              ▷ The main routine of the algorithm
 2:     β ← {∅}, γ ← {∅}, expt_val ← ∅, best_res ← ∅
 3:     Pick at random β ∈ [0, π]ᵖ, γ ∈ [0, 2π]ᵖ
 4:     while (β, γ) can be further optimized, or a limit is reached do
 5:         Initialize res_set ← {∅}
 6:         for a fixed number of shots do
 7:             res_set ← res_set ∪ QAOA(B̂, Ĉ, β, γ, p)
 8:         end for
 9:         From res_set, calculate the expectation value and store in expt_val
10:         Based on the expt_val, pick new 2p angles (β, γ) by classical optimization
11:     end while
12:     From the final res_set, set the result with lowest energy, best_res ← min(res_set)
13:     return best_res
14: end procedure
15: procedure QAOA(B̂, Ĉ, β, γ, t)
16:     Initialize n qubits, |ψ⟩ ← |0⟩⊗ⁿ
17:     Apply Hadamard transform, |ψ⟩ = 1/√2ⁿ(|0⟩ + |1⟩)⊗ⁿ
18:     j ← 1
19:     while j ≤ t do
20:         |ψ⟩ ← e⁻ⁱγⱼĈ|ψ⟩
21:         |ψ⟩ ← e⁻ⁱβⱼB̂|ψ⟩
22:         j ← j + 1
23:     end while
24:     Measure |ψ⟩ in standard basis and store in a classical register o
25:     return o
26: end procedure
```

qubits, can be expressed as:

$$\hat{C} = \sum_a h_a \hat{\sigma}_a^{(z)} + \sum_{a<b} J_{ab} \hat{\sigma}_a^{(z)} \hat{\sigma}_b^{(z)}, \tag{5}$$

$$\text{where } \hat{\sigma}_a^{(z)} = (\otimes_{i=1}^{a-1} \hat{I}) \otimes (\hat{\sigma}^{(z)}) \otimes (\otimes_{i=a+1}^{n} \hat{I}) \tag{6}$$

$$\text{and } \hat{\sigma}^{(z)} = \begin{pmatrix} 1 & 0 \\ 0 & -1 \end{pmatrix}. \tag{7}$$

Here, indices $a, b, i$ label the qubits, $n$ is the total number of qubits, $\hat{\sigma}^{(z)}$ is the Pauli Z operator and $I$ is the identity operator. The other type of Hamiltonian in the QAOA process is a summation of individual Pauli X operators for each qubit involved in the process, which intuitively represents a transverse field in the Ising model:

$$\hat{B} = \sum_a \hat{\sigma}_a^{(x)}, \tag{8}$$

$$\text{where } \hat{\sigma}_a^{(x)} = (\otimes_{i=1}^{a-1} \hat{I}) \otimes (\hat{\sigma}^{(x)}) \otimes (\otimes_{i=a+1}^{n} \hat{I}) \tag{9}$$

$$\text{and } \hat{\sigma}^{(x)} = \begin{pmatrix} 0 & 1 \\ 1 & 0 \end{pmatrix}. \tag{10}$$

In QAOA, the qubits are first put in a uniform superposition over the computational basis states by applying a Hadamard gate, which maps $|0\rangle \rightarrow (|0\rangle + |1\rangle)/\sqrt{2}$, on every qubit. Then, the

Hamiltonian pair $\hat{C}$ and $\hat{B}$ is applied $p$ number of times using a set of angles $\gamma$ and $\beta$, where, for $1 \leq l \leq p$, each $\gamma_l \in [0, 2\pi]$ and $\beta_l \in [0, \pi]$ [1] [4]. The expectation value of the Hamiltonian $\hat{C}$ with respect to the resultant state $|\psi(\gamma, \beta)\rangle$ is calculated as

$$C(\gamma, \beta) = \langle \psi(\gamma, \beta)|\hat{C}|\psi(\gamma, \beta)\rangle. \tag{11}$$

A classical black-box optimizer then uses the expectation value as its input and suggests new $\gamma$ and $\beta$ sets (of length $p$ each). The hope is that as the number of qubits (more specifically, variables) $n$ involved in the optimization increases, if for circuit depth $p \ll n$ we are able to efficiently sample the best solution, we would have an advantage in using QAOA over classical methods. Although Algorithm 2 is a summary of the QAOA method, we recommend readers the original paper [4] for further details.

### 2.1.5 Implicit filtering optimization

As mentioned before, QAOA requires us to give it the sets of angles $\gamma$ and $\beta$ in order to manipulate the state of the quantum system. The most common way to do this is to use classical black-box optimization techniques that do not need the derivative information of the problem [9, 48, 49]. Since the expectation value $C(\gamma, \beta)$ of the objective function cost (or energy) would be approximate in nature, we need an optimization technique that can handle noisy data. The technique of our choice for this work is the implicit filtering algorithm [34].

In essence, implicit filtering or ImFil is a derivative-free, bounded black-box optimization technique that accommodates noise when it tries to suggest the best parameters to minimize the objective function. Various other techniques for noisy optimization exist, such as Bayesian Optimization [50], COMPASS [51], SPSA [52], etc. However, we found implicit filtering the best for our current efforts. For further details, we recommend the book by C.T Kelly on the topic [34].

## 2.2 Related work

One of the first applications of quantum computing for solving problems in the field of linear algebra is the HHL algorithm (named after its creators: Harrow, Hassidim and Lloyd) for solving a system of linear equations [53]. This was followed by works for solving linear least squares [54], preconditioned system of linear equations [55], recommendation systems [56] and many others [57–61]. Although the classical counterparts of the above mentioned algorithms run in polynomial time, the quantum algorithms mentioned above run in the polylog time complexity (provided the problem is well conditioned).

However, there are some caveats with such kind of algorithms [62]. Among the many caveats, we'd like to emphasize on the two that affect the practicality of their utility in the near future. Firstly, they require fault tolerant quantum computers whereas, at the time of writing this paper, we have just entered the NISQ era [5]. Secondly, for the algorithms focused on linear system of equations [53, 55, 57] and least squares [54, 58], the output data is encoded as a normalized vector of a quantum state $|x\rangle$ (which means that the probability amplitudes of the basis states encode the data). This means that we need an efficient method to prepare the input data as a quantum state [62–65]; and the output will be a quantum state as well, which means it wouldn't be available for us in the classical world directly by performing measurement in the standard computational basis. This can be mitigated by either measuring the final state in a basis of our choice if our goal is to know some statistical information about $x$ [53, 66] or learning certain values in $x$ (though that will eliminate the exponential speedup [62]).

---

[1]This is true as long as all possible variable combinations have obj. function costs that are $\geq 1$ in magnitude.

With respect to quantum annealing, O'Malley and Vesselinov's paper in 2016 [27] was one of the first that proposed to solve linear least squares. Other works in this domain were for solving specific NMF problems [17–19], polynomial system of equations [28], underdetermined binary linear systems [67] and polynomial least squares [30]. It's hard to speculate about speedups analytically with (i) D-wave's noisy implementation of quantum annealing [68] and (ii) the problem of exponential gap-closing between the problem Hamiltonian's ground state and its excited states [69]. In the work by O'Malley and Vesselinov [17], they used a time to target benchmark in which classical solvers (Tabu search [70] and Gurobi [71] in their comparison) have to match or find better solutions than the ones returned by quantum annealing (not necessarily the optimal solution) in the same amount of time. The D-wave quantum annealer was able to beat those classical solvers for the benchmark, but the authors also mention that a combination of the two classical techniques would probably perform better than the D-wave by compensating for each other's shortcomings. The other important result in subsequent papers [18, 19] was to show that combining reverse and forward annealing improved results over just using forward annealing for most cases. Golden and O'Malley saw an improvement of 12% over forward annealing [19], but that came at the price of having at least 7 reverse annealing runs per QUBO (which was reported to have the quantum runtime of 29 forward anneals). It is important to note that certain quantum inspired algorithms may perform just as well or better than quantum annealers for such highly dense problems in terms of variable interactions [72]. The above mentioned quantum annealing techniques use the classical Ising model's diagonal Hamiltonian for problem formulation. This means that measuring the post annealing quantum state in computational basis gives us a bitstring which directly encodes the vector $x$ (which we hope to be the best solution to Eqn(1)), unlike a lot of gate-model algorithms like the ones mentioned above [53–55,57,58] that encode the normalized solution vector in the amplitudes of $|x\rangle$. The exception to this rule is the work by An and Lin, where they explore using adiabatic quantum computing, not based on the Ising model, to produce amplitude encoded results for linear system of equations [73].

NISQ-compatible algorithms for efficiently solving linear algebra problems are highly desirable as of the time of writing this paper. The work by Chen et. al [74] proposes a hybrid algorithm that uses quantum random walks for solving a particular type of linear system, producing a classical result in $O(n \log n)$. However, the closest related works to ours are the recent papers that employ variational algorithms [73,75–77]. The major difference however, is that, in those papers: i) The output is encoded as the vector of probability amplitudes of the quantum state $|x\rangle$ and ii) The problems explored thus far are convex in nature and solved in polynomial time classically.

We in this paper implement QAOA on similar problems which were implemented on D-wave's quantum annealer previously, and therefore briefly mention a comparison here. The standard QAOA circuit strategy can be seen as similar to a bang-bang quantum annealing schedule, where cost and driver Hamiltonians are alternated. The quantum alternating operator ansatz extends this with more general operators [78] than available to Ising Hamiltonian annealers. Furthermore, QAOA is a gate-based quantum computational algorithm, a type of framework which promises universal programmability in terms of mapping an arbitrary problem graph to a qubit layout, even if the latter is not all-to-all connected. Conversely, in quantum annealing architectures mapping the logical problem qubits to a graph of physical hardware qubits can be a significant challenge in the general case [79,80]. Our work is certainly not the first in applying QAOA to various relevant computational problems, and we refer the reader to a small list of examples [81–83]; in this work, we make an attempt to highlight some of the salient features and challenges of QAOA in the context of problems applicable to linear algebra and numerical analysis.

# 3 QAOA for BLLS

Before we go deeper, we here set the context of how QAOA will be used in this work. Rather than treating QAOA as an approximation algorithm with theoretical guarantees for the quality of solution obtained, it is used as a heuristic supported by empirical results.

## 3.1 Problem formulation

O'Malley and Vesselinov first gave a QUBO formulation for the BLLS problem [27]. The details of how that is done is in Appendix A. Referring back to Eqn(1), if $A \in \mathbb{R}^{m \times n}$, $b \in \mathbb{R}^m$ and $x \in \{0, 1\}^n$, we can refine Eqn(2) to be

$$F(x) = \sum_j v_j x_j + \sum_{j<k} w_{jk} x_j x_k, \tag{12}$$

$$\text{where } v_j = \sum_i A_{ij}(A_{ij} - 2b_i) \tag{13}$$

$$\text{and } w_{jk} = 2\sum_i A_{ij}A_{ik}. \tag{14}$$

Which means that the number of qubits depends only upon the size of the column vector $x$. All the rows in Matrix $A$ and vector $b$ are preprocessed classically in order to produce the coefficients of the QUBO problem.

By the equivalence stated in Eqn(4), we can then convert the problem into an Ising objective function (plus an offset value, irrelevant for optimization)

$$F(\sigma) = \sum_j h_j \sigma_j + \sum_{j<k} J_{jk} \sigma_j \sigma_k + \text{offset}, \tag{15}$$

$$\text{where } \sigma_j = 2x_j - 1. \tag{16}$$

## 3.2 Mapping to quantum gates

Using the $h$,$J$ coefficients from Eqn(15) along with the mapping to a classical Ising Hamiltonian given in Eqn(6) we get:

$$\hat{C} = \sum_j h_j \hat{\sigma}_j^{(z)} + \sum_{j<k} J_{jk} \hat{\sigma}_j^{(z)} \hat{\sigma}_k^{(z)}. \tag{17}$$

Because the individual components of Eqn(17) commute, we can express the Hamiltonian simulation of $\hat{C}$ with an angle $\gamma_l$ as follows

$$e^{-i\gamma_l \hat{C}} = \prod_j e^{(-ih_j\gamma_l)\hat{\sigma}_j^{(z)}} \prod_{j<k} e^{(-iJ_{jk}\gamma_l)\hat{\sigma}_j^{(z)}\hat{\sigma}_k^{(z)}}. \tag{18}$$

Similarly, the exponential of hamiltonian $B$ can be broken down as

$$e^{-i\beta_l \hat{B}} = \prod_j e^{(-i\beta_l)\hat{\sigma}_j^{(x)}}. \tag{19}$$

In order to realize Eqn(18) and Eqn(19), we use the following gates

$$R_x(\omega) = e^{-i\frac{\omega}{2}\hat{\sigma}^{(x)}} = \begin{pmatrix} \cos \omega/2 & -i \sin \omega/2 \\ -i \sin \omega/2 & \cos \omega/2 \end{pmatrix}, \tag{20}$$

$$R_z(\omega) = e^{-i\frac{\omega}{2}\hat{\sigma}^{(z)}} = \begin{pmatrix} e^{-i\omega/2} & 0 \\ 0 & e^{i\omega/2} \end{pmatrix}, \tag{21}$$

$$\text{CNOT} = \begin{pmatrix} 1 & 0 & 0 & 0 \\ 0 & 1 & 0 & 0 \\ 0 & 0 & 0 & 1 \\ 0 & 0 & 1 & 0 \end{pmatrix}. \tag{22}$$

While Eqn(20) is the only gate needed to realize Eqn(19), Eqn (21) alone can merely help with the single qubit components of Eqn(18). For the components that require two qubit interaction, the following gate combination (expressed diagrammatically) is used as a template

$$e^{(-iJ_{1,2}\gamma_l)\hat{\sigma}_1^{(z)}\hat{\sigma}_2^{(z)}} = \tag{23}$$

While Eqn(23) shows the ZZ interactions for adjacent qubits, this strategy can be generalized to any pair of qubits in the system. Appendix B provides an example of a QAOA circuit for BLLS.

### 3.2.1 For IBM Q specific gates

Our experiments were done on IBM Q devices (`ibmq_london` and `ibmq_athens`) available to us through the IBM Q Network. The former has the following basis gates $\{U_1, U_2, U_3, \text{CNOT}, I\}$ and the latter has $\{CX, ID, RZ, SX, X\}$. Although most of these gates are defined and established in the literature well [1], we need to elaborate more on $U1, U2$ and $U3$ as they are IBM Q specific gates.

$$U_1(\lambda) = \begin{pmatrix} 1 & 0 \\ 0 & e^{i\lambda} \end{pmatrix}, \tag{24}$$

$$U_2(\lambda, \phi) = \begin{pmatrix} 1/\sqrt{2} & -e^{i\lambda}/\sqrt{2} \\ e^{i\phi}/\sqrt{2} & e^{i(\lambda+\phi)}/\sqrt{2} \end{pmatrix}, \tag{25}$$

$$U_3(\theta, \phi, \lambda) = \begin{pmatrix} \cos \theta/2 & -e^{i\lambda} \sin \theta/2 \\ e^{i\phi} \sin \theta/2 & e^{i(\lambda+\phi)} \cos \theta/2 \end{pmatrix}. \tag{26}$$

We can implement Eqn(20) and Eqn(21) [84] as

$$R_x(\omega) = U_3(\omega, -\frac{\pi}{2}, \frac{\pi}{2}), \tag{27}$$

$$R_z(\omega) = U_3(\pi, 0, \pi) U_1(-\frac{\omega}{2}) U_3(\pi, 0, \pi) U_1(\frac{\omega}{2}). \tag{28}$$

Another practical consideration to be taken is the qubit connectivity of a real quantum computer. As the number of qubits increase, it is safe to assume that full connectivity between physical qubits is not feasible to engineer. This means that for distant-qubits to interact with each other, we would need logical qubit replacement using SWAP operations. Appendix C elaborates on this with a demonstration with IBM Q gates.

## 3.3 Experimental methods

The dataset used in our experiments was randomly generated (seeded for reproducibility) consisting of $A \in \mathbb{R}^{40 \times n}$, $b \in \mathbb{R}^{40}$ and $x \in \{0,1\}^n$ where $n \in \{3,4,5,9,10\}$ is the size of the problem. Due to the exponential nature of simulating quantum computation on classical hardware and the limited resources in our hands, we decided to limit our problem sizes to 10 qubits. Instead of allocating computational resources to even larger qubit numbers, we focus on: i) increasing the number of problem instances we average over, ii) comparing exact wavefunction versus bitstring-sampling experiments, iii) comparing QAOA performance (waveform) with simulated annealing and random sampling. All values for $A$ in the dataset are generated by uniformly sampling floating point approximations of real values in the interval $[-1.0, 1.0]$, and then rounding the values to 3 decimal places. For each value of n, we generate 100 test cases with 40 cases in which $Ax^* = b$, where the best solution $x^*$ is sampled randomly and $b \leftarrow Ax^*$. The other 60 cases have $Ax^* \neq b$, where $b$ is generated similarly to $A$ and the best solution $x^*$ is found by going through all $2^n$ possible values for $x$. This is done to cover both scenarios of the least squares problem. The matrix $A$ is a sparse matrix having density of 0.2, this was done because sparse matrices have a lot of applications in numerical computation and machine learning [85–87].

We use the QISKIT [88] SDK to write our own implementation of the QAOA algorithm. As mentioned before, ImFil [34] is our black-box optimizer of choice. The only parameter of ImFil we choose to control here is the budget, which governs the maximum iteration limit. The rest of the ImFil parameters for our experiments use their default values. Similarly, unless explicitly stated, all qiskit parameters values taken are default as well. In the later sections, the term quantum virtual machine (QVM) is used to denote classical simulators broadly (unless specified to be of a particular kind). Similarly, the term quantum processing unit (QPU) is used to denote a quantum device. All classical simulations were conducted on standard `x86-64` based laptops. Following is a list of the main experiments we conducted, with a few slightly-modified ones described later on in the results section as applicable.

### 3.3.1 Experiments with no noise

Our first set of experiments on the dataset were done on a simulator with the statevector QVM, giving us the exact waveform. This means that we are able to compute the exact expectation value $C(\gamma, \beta)$ for the set of angles $\gamma$ and $\beta$. These experiments help assess the performance of QAOA in a perfectly noiseless environment for a large dataset.

The above set of experiments were done for $p =$1, 2 and 3 with random starting points: 20 for $p = 1$, 40 for 2 and 60 for 3 (seeded for reproducibility). Our preliminary study suggested a budget of 200 iterations for $p = 1, 2$ and 400 iterations for $p = 3$ respectively. This ensured that at least 70% of our tests converged within the budget while being computationally feasible. At the end of the process, for each problem, the best result from all the starting points is chosen and recorded.

### 3.3.2 Experiments to compare no noise and shot-noise performance

For our next set of experiments, we use measurement based results on the simulator. Each circuit is run a number of times, specified by the 'shots' parameter. This means that the expectation value we get for a given $\gamma$ and $\beta$ is approximate in nature. Thus, while quantum circuit simulation itself is noiseless and deterministic in producing the same wavefunction before taking each shot, a finite number of shots is sampled from the resulting wavefunction output probability distribution, introducing a stochastic component. In a real quantum device, one is always limited to this finite tomography, as one has no direct access to the qubit register's quantum wavefunction.

Since in a real quantum device, we do not have access to the qubit register's waveform, simulations with shot-noise are important to conduct. Each experiment was done 10 times per shot value. The shot values chosen for these experiments are in the set $\{2^i | n-2 \le i \le n+2, i \in \mathbb{Z}\}$. We chose this range in order to observe the performance in the limit of perfectly reproducing the wavefunction.

Also, the problem instances chosen for this set of experiments are a random subset of the original dataset. For each problem size $n \in \{3, 4, 5, 9, 10\}$, we randomly choose 5 problems of the 100 problems (while maintaining the $2:3$ ratio of the problems by their type). This is done because doing the shot-noise experiments on the original dataset would be computationally infeasible for the limited computational resources at our disposal, since each shot-noise experiment is at least 50 times slower than its statevector counterpart.

The parameters of these experiments have also been modified accordingly. They were done for $p =1$, 2 and 3, for a budget of 200 iterations with random starting points: 5 for $p =1$, 10 for 2 and 15 for 3 (seeded for reproducibility), with the best result being chosen and recorded. For fair comparison, this subset of problem instances was also run with the statevector QVM for the same parameters.

### 3.3.3 Experiments on an IBM Q device

Based on the results of the first two sets of experiments, we design our experiments for the 5 qubit IBM Q devices 'ibmq_london' and 'ibmq_athens'. In a real device like this one, the qubits face decoherence issues, coherent gate errors, control errors, incoherent gate errors, leakage, cross-talk, readout noise and more. The first set of IBM Q experiments was to run QAOA for problems with $n =5$, for parameters $p =1$, budget of 200 iterations and a shot value of 1024. The reason for choosing these parameters for QAOA is to take into account the gate depth limitation and noisy computation, thus choosing the minimal number of qubits while still covering a non-trivial problem graph structure, which can still be easily verified with classical methods at this size. The next set of experiments was to take the $\gamma$ and $\beta$ from the results of the statevector experiments done in Section 3.3.2 (where $n = 5$) and to try and recreate the distribution and expectation values using the quantum computer.

## 3.4 Results

Two metrics we use here to quantify the performance of QAOA in the simulations are (i) the probability of sampling the best possible solution (or the ground state of Hamiltonian $\hat{C}$) and (ii) the relative error $\varepsilon_r$ of the expectation value $C(\gamma, \beta)$ (from Eqn(11)) with respect to the ground state energy $E_{gs}$. We define $\varepsilon_r$ as

$$\varepsilon_r = \left| \frac{C(\gamma, \beta) - E_{gs}}{E_{gs}} \right|. \tag{29}$$

We note that $\varepsilon_r$ is zero when the expectation value is exactly equal to the groundstate energy, and can grow beyond 1 depending on the spectral width as compared to the groundstate cost.

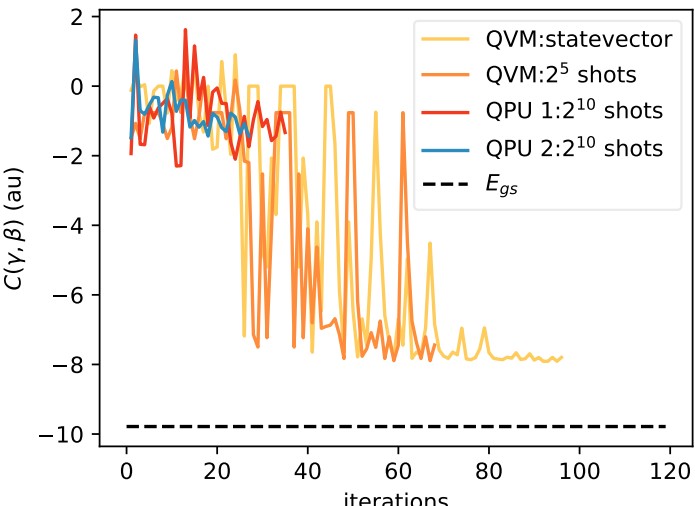

Figure 1: One instance of convergence behaviour for the QAOA optimization routines run on an exact statevector simulator, shot-noise simulator with $2^5 = 32$ shots, and two 5-qubit IBM Q processors: `ibmq_london` (QPU 1) and `ibmq_athens` (QPU 2) with $2^{10} = 1024$ shots each. The problem in question has 5 binary variables mapped to 5 qubits and is done for a QAOA depth of $p = 1$. All $C(\gamma, \beta)$ values dealing with shot-noise (QVM and QPU) are approximate in nature. For comparison, we include the exact ground state energy. The data in the figure is a result of the experiments described in Sections 3.3.2 and 3.3.3.

### 3.4.1 Optimization trajectory for QAOA

In Figure 1, we see an example of how QAOA with ImFil performs on a BLLS problem. As the iterations progress, the fluctuations in the energy expectation value also reduces. This happens either till the black box optimizer converges to a solution (depending on default internal parameters in our case) or the iterations have reached the maximum threshold (governed by the budget). Here the experiments done with the statevector QVM, which has access to the exact energy expectation value, sets the baseline for the other modes of experiments. One interesting thing you will see is that the statevector QVM can not always converge to the ground state energy $E_{gs}$. This is because the results are computed for QAOA depth $p = 1$. For low circuit depth $p$, no guarantees are known to converge to the exact groundstate. As an upper bound, Farhi's theorem [4] states that $C(\gamma, \beta)$ can reach $E_{gs}$ as $p \to \infty$, in the general case, assuming the optimal circuit angles are known. While our simulations containing shot-noise due to measurement do relatively well against statevector results, the experiments on a real quantum device are mixed. At the time of writing, the IBM Q devices we tested on did not approximate the theoretically-optimal QAOA result distribution very well, but it still finds the best solution every time. This means they failed to converge using expectation values $C(\gamma, \beta)$ they produced, but randomly happened to sample the ground state during the process. We have discussed this further in Section 3.4.7 and have included the results for reference.

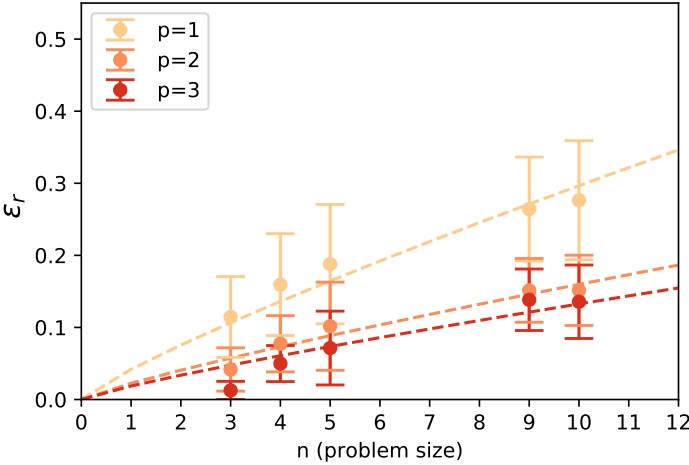

Figure 2: Comparison of converged final relative error $\varepsilon_r$ (median), for $p \in \{1, 2, 3\}$, as a function of problem size $n \in \{3, 4, 5, 9, 10\}$. Simulations performed with access to the exact wave form (statevector QVM) done on 100 problem instances per $n$ with an optimization budget of 200 iterations for $p \in \{1, 2\}$ and 400 iterations for $p = 3$. The dashed curves in the plot are fitted to the experimental data. The information presented in this figure is the output of experiments described in Section 3.3.1.

### 3.4.2   QAOA results with no noise

Before we study the results of QAOA for BLLS with shot-noise, it is important to evaluate the theoretical performance of the same without any noise at all. Figure 2 shows the relative error growth with respect to the problem size $n$ for the experiments described in Section 3.3.1. We use median as measure of central tendency and median absolute deviation (MAD) for our error bars. Simulations larger than $p = 3$ take a lot more time for the complete dataset and were computationally infeasible for this project.

Using the information from the medians calculated, we attempt to fit curves for different values of $p$. Based on the curve fit, we find that the growth in $\varepsilon_r$ to be polynomial in nature (as described by $\varepsilon_r = a \times n^b$, where $a$ and $b$ are coefficients, see Appendix E for details). This polynomial growth may be partly attributed to some spin configurations with energies close to the ground state. You can see that going from $p = 1$ to $p = 2$ decreases the relative error moderately. The difference in performance between $p = 2$ and $p = 3$ is more modest, particularly for the larger problem sizes $n$. There may be room for further improvement if we allow a larger simulation time budget, for example by tightening the classical optimizer's convergence parameters and increasing the number of initial starting points for the optimizer. Further rigorous experimentation would be required to draw definite conclusions about the scaling based on such numerics.

### 3.4.3   No noise vs shot-noise optimization

Figure 3 shows us how QAOA with ImFil performs for the parameters described in Section 3.3.2 for statevector and measurement based results for $2^{n+2}$ shots. We have 5 different problem instances per $n \in \{3, 4, 5, 9, 10\}$. The reason for choosing $2^{n+2}$ shots for this comparison was to try and see if the optimizer could replicate the statevector results given plentiful shots. In subsequent figures, we'll be showing the performance of the optimizer with fewer number of shots.

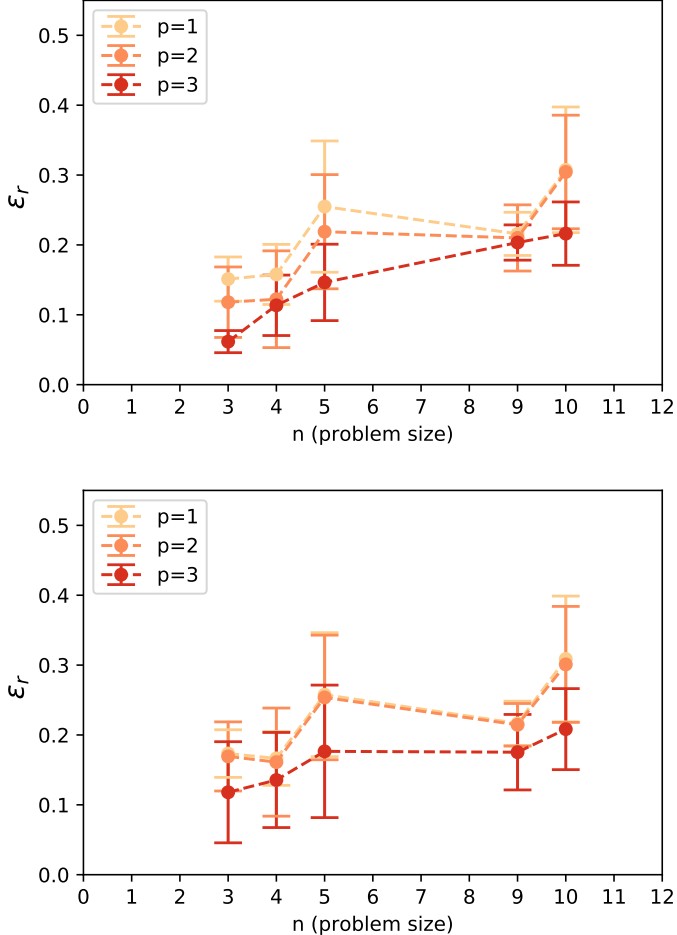

Figure 3: Comparison of converged final relative error (median), for $p \in \{1, 2, 3\}$, as a function of problem size $n \in \{3, 4, 5, 9, 10\}$, done on 5 problem instances per $n$ with a budget of 200 iterations. (TOP) shows the results from the optimization having access to the exact waveform (statevector simulator), while (BOTTOM) shows the results using a shot-noise simulator with $2^{n+2}$ shots. For drawing the bottom plot, we use the best angles found using shot-based optimization and used the statevector QVM for one more run at those angles in order to compute the exact expectation value and corresponding relative error. These results are from the experiments described in Section 3.3.2.

We can see the similarities between the top and bottom plots in Figure 3. The main difference however, seems to be the result and error bar overlap between the results of $p = 1$ and 2. While the two lines are close to each other in the statevector results uptil problem size of 9, the measurement-based results for the two parameters are extremely close to each other (when considering median and MAD). This could be attributed to the noise due to approximate results, or due to the small number of experiments we average over, as detailed in Section 3.3.2.

Another effect of the smaller dataset here is that the relative error's growth doesn't seem fully monotonic to the problem size, unlike in Section 3.4.2. However, it still shows a general upward trajectory. Finding optimal parameters for $p = 4$ and upwards become computationally infeasible due to the time required, even for the smaller dataset we worked with in Figure 3. This can be attributed to the search space growing rapidly as $p$ is increased [4].

### 3.4.4 Probability of sampling the ground state

We compare the probability of sampling the ground state of the BLLS problem (henceforth also referred to as the success probability) for QAOA with two classical methods: random sampling and simulated annealing. For the simulated annealing experiments, we chose 10 steps, an exponential decay schedule, and ran it on problem sizes $n$ from 3 to 20 (with 100 problems per n). The choice of 10 steps was made in order to make a reasonable comparison against our QAOA results, given the size of the problem and the QAOA depths considered. For this problem size, simulated annealing is very cheap computationally on a classical computer, as compared to simulating the quantum circuit with a classical computer. This allows us to consider statistics of a larger number of random problems per $n$. See Appendix E for further details.

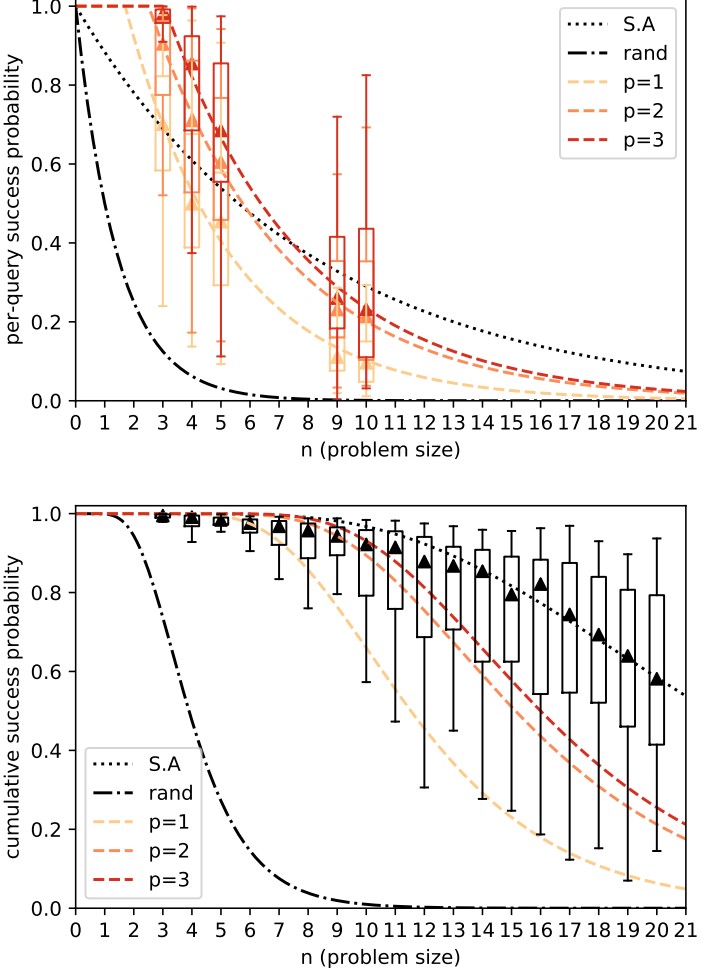

Figure 4: Comparison of success probabilities of sampling the ground state, for QAOA at depths $p = 1, 2, 3$, simulated annealing and random sampling. (TOP) shows the success probability per-query and (BOTTOM) shows the cumulative success probability after 10 queries. The box plots indicate the nature of the experimental results (with medians as triangles) while curves in the figure are fitted to them. The QAOA results are from the statevector (no noise) experiments as described in Section 3.3.1 and the approach for comparing it with simulated annealing and random sampling is described in Section 3.4.4.

The success probabilities for QAOA are calculated from the outputs of the statevector experiments described in Section 3.3.1. More specifically, these would be the success probabilities from sampling the optimized waveform $|\psi(\gamma, \beta)\rangle$.

While the success probability of uniform random sampling is easy to calculate per query as $1/2^n$, things are not as straightforward with simulated annealing. The main reason being that simulated annealing requires $k$ number of steps to arrive at a result. Assigning $k = 1$ would be equivalent to random sampling, but by this logic, assigning $k > 1$ would yield us a cumulative success probability for random sampling.

Thus, our proposed comparison method assumes an effective probability $\mathcal{P}_{\text{eff}}$ which is the success probability per query for simulated annealing. This $\mathcal{P}_{\text{eff}}$ is not directly observed, but calculated by extrapolation from simulated annealing run on $k > 1$ steps. After $k$ steps, the cumulative success probability would be:

$$\text{cumulative success prob.} = 1 - (1 - \mathcal{P}_{\text{eff}})^k. \tag{30}$$

Doing a curve fit modeled on Eqn(30) for the results of simulated annealing with $k = 10$ steps yields us the value of $\mathcal{P}_{\text{eff}}$ which would be the effective success probability, for simulated annealing, per query. Conversely, curve fits were done on the QAOA per query success probabilities and were extrapolated to cumulative probabilities for 10 queries. This can be conceptualized as the success probability of measuring the ground state at least once from an optimized QAOA quantum state when prepared and measured 10 times. The resulting information is plotted in Figure 4 along with the box plots from the experiments. We refer the interested reader to Appendix E for the details.

As Figure 4 suggests, the success probability for all methods decrease exponentially as the size of the problem increases. The comparison of QAOA with uniform random sampling for BLLS corroborates with previous work done on applying QAOA to a different problem [9]. Even with $p = 1$, QAOA performs much better than uniform random sampling. Simulated annealing on BLLS however, yields a higher success probability than QAOA at depth $p = 3$ when $n > 8$ (for 10 steps). This empirical result is supported by the theoretical work on QAOA for MAXCUT [89] which indicates that there exists classes of graphs for which quantum advantage is not possible for $p < 6$. Another work focusing on numerical simulations on MAXCUT problems suggests any QAOA advantage would need quantum computers to have in the order of hundreds of qubits at the very least [90]. That being said, while our result is interesting, it is not conclusive on its own. As future work, we recommend testing on larger problems, for higher $p$ and different simulated annealing steps and schedules.

It is one thing to calculate probability, it is another to sample the best solution (or ground state) from a quantum state after QAOA. Figure 5 displays the number of experimental instances where we sample the ground state, at least once, for a particular set of parameters, across various problem sizes and instances (for shot-noise experiments in Section 3.3.2). We contrast this with the analytical results of getting the ground state by uniform random sampling. Here, we see that for optimization done with upto $2^n$ shots, QAOA has a clear advantage over random sampling. This can be explained by the mechanism of QAOA, which selectively amplifies those bitstring sampling probabilities which have the lowest energy, while suppressing those with higher energy. In this way, the success probabilities may be greatly enhanced over the naive random sampling from the uniform probability distribution.

### 3.4.5 Effect of noise on expectation value

While Figure 1 indicates to us that the current generation of devices are unable to do the entire variational optimization procedure for this problem very effectively, studying the effects of noise on approximating $C(\gamma, \beta)$ for a particular set of angles gives us information that may

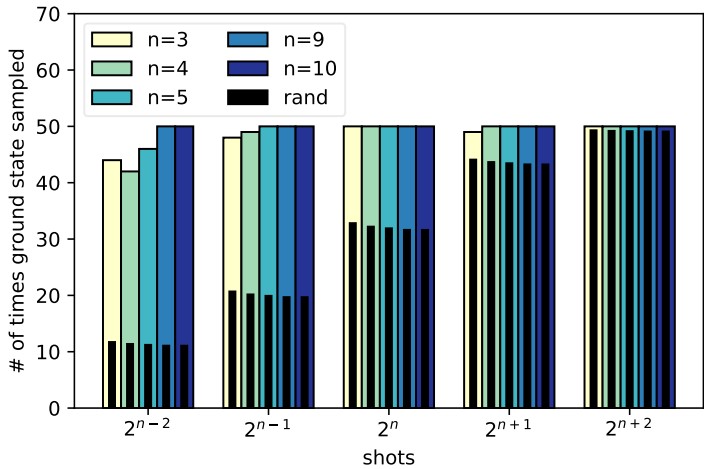

Figure 5: In this bar graph, we collect the number of experiment instances in which we observe the exact ground state bitstring, at least once (on the Y axis) for $n \in \{3, 4, 5, 9, 10\}$. Each $n$ has 5 problem instances, which is repeated 10 times for a given shot value. We compare the QAOA shot-noise simulator experiments (color-coded with the number of binary variables, $n$), with the results one would expect randomly sampling from a uniform distribution *shots* times (black bars, labeled with rand, x-axis positioning corresponding to its colour-labeled partner). The QAOA data in this figure comes from sampling the circuit with optimized angle sets $\gamma^*$ and $\beta^*$, after acquiring them by optimizing for a QAOA depth of $p = 3$ and a budget of 200 iterations, as described in Section 3.3.2.

be useful for future research and development. Here, we investigate the impact of noise by approximating a problem's expectation value on a noiseless waveform simulator (statevector QVM), a quantum device (QPU), and a range of increasingly noisy simulations in between the two.

In order to do this, we first need a template for the noise model that we can use to make QVMs (for noisy classical simulations). What IBM Q allows through Qiskit [88] is to produce a simplified approximate noise model based on the properties of a real QPU (`ibmq_athens` in our case). We take this extracted noise model as our template and scale each of the quantum and readout errors down in powers of two (upto $2^{-7}$ for our experiment). Then, from our problem set of 5 variables, we randomly choose 30 problems where the ground state bitstrings do not have all zeros or ones in them. The reason being that all-zeros or all-ones bitstrings might be comparatively easier to sample from a QAOA circuit. For our selected problems, we take the $(\gamma, \beta)$ pairs responsible for producing the best $C(\gamma, \beta)$ in their noiseless waveform simulations (circuit depth of $p = 1$), and use them with the various scaled noise models and the device to approximate the expectation value. In order to reduce the effect of shot-noise, we calculated the expectation value with 8192 shots. Since actual devices are limited by qubit connectivity, this also needs to factor in our noisy simulations as it increases the gate operations and the overall circuit depth (see Section 3.2.1 and Appendix C). This information about how physical qubits are connected is referred to as a coupling map of a device. For our experiment, all noisy simulations were done either assuming all-to-all connectivity, or with the device limited-coupling map restriction.

Figure 6 plots the relative errors of $C(\gamma, \beta)$ (with respect to the problem ground state) for the noiseless waveform, quantum device and noisy simulation expectation values that interpolate between the two. Though our approximation of the device noise model often does not

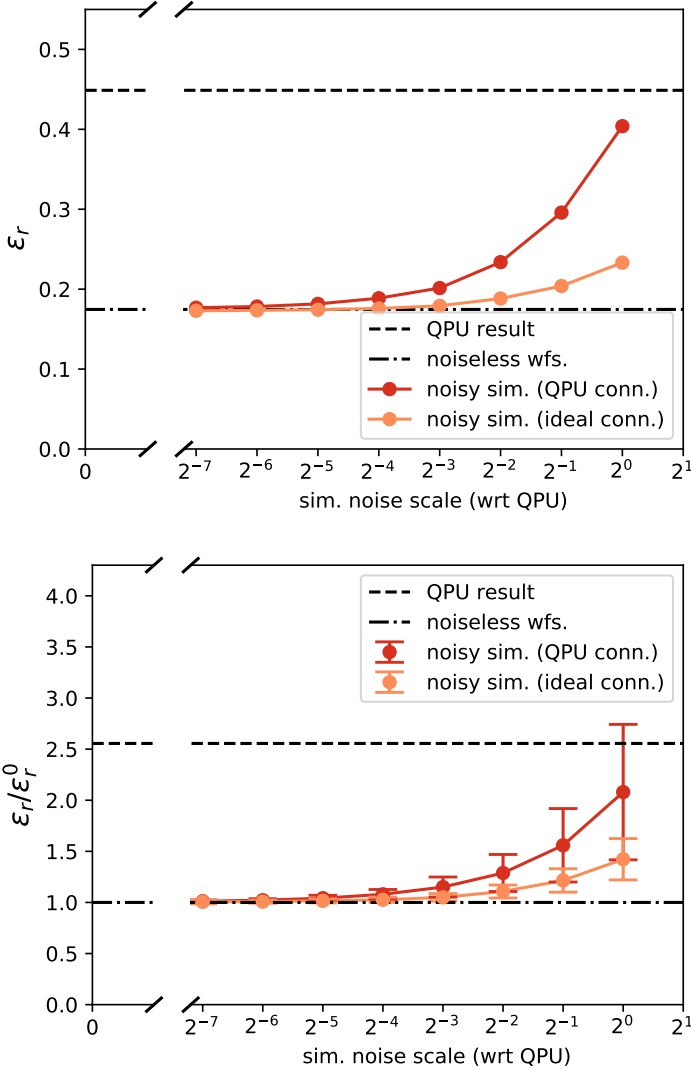

Figure 6: (TOP) Comparison of relative error $\varepsilon_r$ with respect to the scaled noise model simulations for a typical 5 qubit problem instance (log scale). Noiseless waveform simulation as well as QPU results are also plotted for reference. (BOTTOM) Relative error factor $\varepsilon_r/\varepsilon_r^0$ with respect to the scaled noise model simulations (log scale). Here, $\varepsilon_r^0$ refers to the relative error observed in the case of the noiseless wavefunction simulator and $\varepsilon_r/\varepsilon_r^0$ measures $\varepsilon_r$ as a factor of $\varepsilon_r^0$. Values plotted on the y axis are medians (error bars are in Median Absolution Deviations) calculated from the values of the 30 selected 5 qubit problems. The QPU result in the bottom plot has a MAD of 1.04. Both plots are made with results from a QAOA depth of $p = 1$, for device-limited coupling map restrictions (QPU conn.) and all-to-all connectivity (ideal conn.).

mimic the exact effect of the device noise on expectation values, for the scope of this work, it serves its purpose well since the true device result and the unscaled noise model results are not wildly divergent. From the figure, we see that for this type of a problem (and parameters), the relative error in $C(\gamma, \beta)$ that we observe while using the device is $\approx 2.5$ times the relative error of waveform simulations, statistically speaking. An in-depth study on the effect of various noise types would be required to make a definite statement on the effect of noise

on QAOA (as it relates to this problem).

Another observation we can make is the how an ideal coupling map, where every qubit is connected to every other qubit, gives us significantly better results than the device's qubit map, for the very same noise model. This is mainly due to the additional SWAP gates needed for each qubit to interact with all the other qubits for the latter.

### 3.4.6 Effect of shot number on optimization

For QAOA to become practical, the shot number chosen for the computation has to be far less than the number of eigenstates for our cost Hamiltonian ($2^n$ for our case). For this work, we chose not to randomly guess a shot number value but rather get an understanding of the optimization performance for a set of shot numbers in $\{2^i | n - 2 \leq i \leq n + 2, i \in \mathbb{Z}\}$. We hope this helps all future research work in finding better estimates for the least amount of shots required for QAOA, especially for these type of applications. In Figure 7 we see the optimization result for a problem instance where $n = 5$. The shot optimization is compared with noiseless statevector optimization. Here it is important to point that we optimized using the stochastic blackbox (using a given shot number) and then calculate the exact expectation value using the wavefunction (i.e, with the statevector QVM) in order to assess the true value of relative error. For the most part, our experiments show that as the problem size increases, we see the optimizer do well even with $2^{n-2}$ shots. This seems to indicate that the number of shots required to get a good optimization may not be exponential in terms of the problem size, or at least with a smaller exponent than applying random sampling from a uniform distribution. Further research is needed.

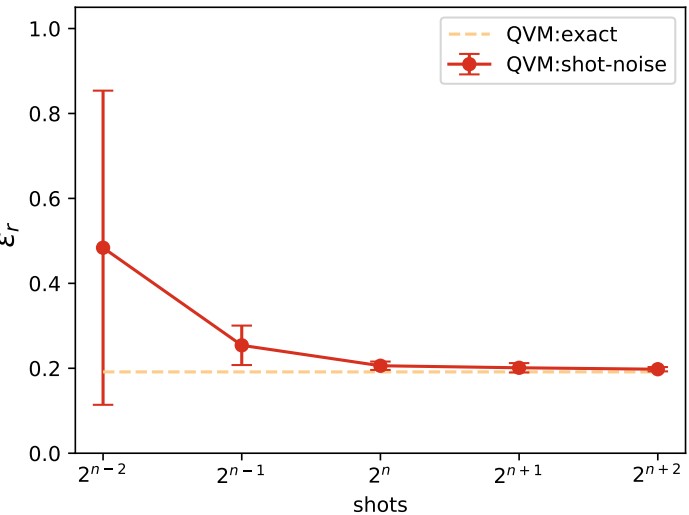

Figure 7: Relative error of the optimized QAOA output distribution (y-axis), as a function of number of shots (x-axis), for a problem instance of the $n = 5$ case. Circles indicate median, and we plot the error-bars (using MAD) when approximating the expectation value with a finite number of shots. QVM:shot-noise refers to the values after the shot-noise optimization and QVM:exact denotes the relative error found after noiseless statevector optimization and is $\approx 0.2$. The QAOA depth for all values in the plot is $p = 1$. Data represented in the figure is an output of an experiment done in Section 3.3.2.

### 3.4.7 IBM Q device performance

We briefly mentioned our results on the quantum processors in Section 3.4.1. The good news is that the IBM Q devices were always able to find the best solution for our optimization experiments. But that came at the price of taking 1024 shots for each QAOA iteration, which is relatively expensive for a problem size of 5 qubits. When we lowered the shot number, the optimal bitstring was not always sampled and the convergence deteriorated further. The immediate cause of why the optimization process on the device was not close to the simulation results, is the inability of the device to approximate the distribution of the measured bitstrings (for a given circuit).

We shall use the quantum volume (QV) [91] measure for comparing the devices we ran experiments on. In short, it is a scalar number which quantifies the largest square quantum circuit (depth = width) that the QPU can successfully implement. When we consider a pair of $(\gamma, \beta)$ angles obtained after noiseless optimization; `ibmq_athens`, a device with a QV of 32, comes closer to the actual probability distribution compared to the distribution produced by `ibmq_london` with a QV of 16. For reference, we provide an example of this in Appendix D for the readers. However, even the QPU with QV of 32 wasn't able to efficiently do its part in the variational procedure. And while this result is not surprising for QAOA on the current generation of cloud accessible quantum computing devices [92–95], it is important to emphasize on it particularly for readers from other domains who may be interested in QAOA for their own applications.

It is important to take into account the entire context here. Firstly, the problem graph of our use case is fully connected. Due to the sparse connectivity of the on-chip qubits in the device, logical qubits have to be swapped around a number of times for them to be able to entangle with each other (for the $ZZ$ interactions of our problem). This makes the median gate depth for the final (transpiled) circuits to be about 42 for `ibmq_london` (t-shaped coupling map) and 38 for `ibmq_athens` (linear coupling map). Since two qubit gate fidelity is still low (at the time of writing), the error propagates across the circuit. Secondly, due to the large circuit depth on the real device, we need to take decoherence into account. Thirdly, readout-errors were not considered here and they have significant impact on the noise in the qubit measurement results. It should also be emphasized that in this work, we primarily focused on how to model the BLLS problem using QAOA. Thus, the experiments on the real devices were done "as is", in order to demonstrate the near-term implementability, without any error mitigation [96]. This could be looked at for future work.

## 3.5 Discussion

We can see the various possibilities and potential advantages QAOA may provide in solving BLLS and similar problems. However, there are challenges that need to be addressed. These are both theoretical and practical in nature.

One theoretical challenge is the proper pre-processing of the problem Hamiltonian by scaling and shifting the coefficients of the objective function, such that we optimally make use of the parameter space $\beta_l \in [0, \pi], \gamma_l \in [0, 2\pi]$ (most of the problems in the dataset did not suffer from this issue, as we found the default scaling to work well already). However, scaling the problem way beyond necessity also creates issues as the energy landscape is periodic in nature [4]. Thus, one possible way is to use scaling as a heuristic within the QAOA process, and treat it as a hyperparameter to optimize over.

Another challenge, which is both theoretical and practical in nature, is the full connectivity in our problem and in most hard optimization problems in general [97]. Computationally non-trivial problems typically require a high degree of graph connectivity (for instance, planar graphs are easy to solve classically [98]). Simultaneously, a high connectivity poses a challenge

in quantum chip implementation because not all gatesets implement non-nearest neighbour interactions natively. Those need then be implemented effectively by means of a swap network approach [95]. For future work, one option is to modify the problem formulation by not considering the ZZ interactions of a pair of qubits, if its coefficient's magnitude falls below a user defined threshold. This can potentially make it easier for QAOA to run, but its effectiveness in finding the ground state would come under scrutiny. Nonetheless, it can make for interesting future work. Also, error mitigation techniques and readout error correction will also help in improving the results [95]. It would take a combination of the above mentioned approaches to improve performance on a real device.

Challenges aside, we observe the effects of noise on producing the expectation value for a set of $(\gamma, \beta)$ angles. Although we see that a device with QV of 32 is able to do a closer approximation of the expectation value (for QAOA depth of $p = 1$) than a device with a QV of 16, it is still not good enough for the variational optimization procedure. We also see the potential of a better coupling map for significantly improving the results for the very same noise model. Thus, the solution for improving the results for problems in this application domain need not be considered in unidimensional terms (improving the noise only or coupling only).

One of the next steps would be to explore if QAOA can be valuable in applications that require BLLS as a subroutine, such as NBMF [17]. Another step can be to try the BLLS problem on other types of quantum computers [9–13] to see how different hardware implementations fare. Finally, from a practical standpoint, it may also be beneficial to apply a modified QAOA-like ansatz based on just nearest-neighbor connected CNOT gates, along with a larger number of parameters [99, 100], or apply a SWAP-network QAOA approach [95]. This may potentially lead to faster convergence and better QPU performance for architectures with limited connectivity.

# 4 Conclusion

In this work, we described the implementation of a binary optimization problem, relevant to hard problems in linear algebra, on a gate-based quantum computer via a QAOA approach suitable for NISQ devices. In our simulations, we show how simulated annealing with small number of steps can outperform QAOA for $p \leq 3$ in terms of sampling the ground state for BLLS type problems. We show that the ImFil optimizer backend performs well under shot-noise for this problem type. From our experiments on IBM Q cloud-based quantum processors and simulations with noise models, we conclude that although machines with a Quantum Volume of 32 are starting to better approximate the noiseless simulation results, it is still challenging to implement linear-depth, high-connectivity circuits on the latest hardware available when it comes to variational optimization for this type of a problem. We expect a future experimental implementation would benefit greatly from gate-error mitigation techniques and post-processing readout errors. It would furthermore be very interesting to see what other hard problems in linear algebra may be implemented using the QAOA ansatz and what their expected performance would be.

# Acknowledgments

The authors would like to thank the people who made this collaboration possible. From UMBC: Dean Drake, Wendy Martin and Cameron McAdams from the office of the vice president for research (OVPR), the office of technology development (OTD) and the office of sponsored programs (OSP) respectively. From Qu & Co: we'd like to thank Benno Broer, CEO and co-

founder of Qu & Co. This work was performed in part using IBM Quantum systems as part of the IBM Q Network.

## A  Detailed QUBO formulation for binary linear least squares

In this section of the Appendix, we describe the method by which O'Malley and Vesselinov [27] formulated the binary linear least squares (BLLS) problem. This QUBO formulation will be converted into its equivalent Ising objective function and used in QAOA. Let us begin by writing out $Ax - b$ which would help us in minimizing $x$ and thereby solve Eqn(1)

$$Ax - b = \begin{pmatrix} A_{11} & A_{12} & ... & A_{1n} \\ A_{21} & A_{22} & ... & A_{2n} \\ \vdots & \vdots & \vdots & \vdots \\ A_{m1} & A_{m2} & ... & A_{mn} \end{pmatrix} \begin{pmatrix} x_1 \\ x_2 \\ \vdots \\ x_n \end{pmatrix} - \begin{pmatrix} b_1 \\ b_2 \\ \vdots \\ b_m \end{pmatrix}, \tag{A.1}$$

$$Ax - b = \begin{pmatrix} A_{11}x_1 + A_{12}x_2 + ... + A_{1n}x_n - b_1 \\ A_{21}x_1 + A_{22}x_2 + ... + A_{2n}x_n - b_2 \\ \vdots \\ A_{m1}x_1 + A_{m2}x_2 + ... + A_{mn}x_n - b_m \end{pmatrix}. \tag{A.2}$$

Taking the 2 norm square of the resultant vector of Eqn(A.2), we get

$$\|Ax - b\|_2^2 = \sum_{i=1}^{m}(|A_{i1}x_1 + A_{i2}x_2 + ... + A_{in}x_n - b_i|)^2. \tag{A.3}$$

Because we are dealing with real numbers for $A$ and $b$, $(|.|)^2 = (.)^2$

$$\|Ax - b\|_2^2 = \sum_{i=1}^{m}(A_{i1}x_1 + A_{i2}x_2 + ... + A_{in}x_n - b_i)^2. \tag{A.4}$$

And thus, the coefficients in Eqn(2) are found by expanding Eqn(A.4), which gives us Eqn(13) and Eqn(14). You will notice that there is a constant value from Eqn(A.4) that we leave out of Eqn(13) and Eqn(14). Because this value is not a coefficient for any of the variables, we can't optimize over it and it's left as is, which is $\|b\|_2^2$. Also, the ground state energy (QUBO) for when $\|Ax^* - b\|_2 = 0$ where $x^*$ is the best solution, is $-\|b\|_2^2$.

## B  Example QAOA circuit for BLLS

Let us consider a simple problem, without loss of generality, to demonstrate how quantum circuits for BLLS can be designed. Consider the following problem, Find $x \in \{0, 1\}^3$ such that $\|Ax - b\|$ is minimized, where

$$A = \begin{pmatrix} 2 & 1 & 1 \\ -1 & 1 & -1 \\ 1 & 2 & 3 \end{pmatrix} \tag{B.1}$$

$$\text{and } b = \begin{pmatrix} 3 \\ 0 \\ 3 \end{pmatrix}. \tag{B.2}$$

This particular problem is more appropriately categorized as a linear system of equations $(A \in \mathbb{R}^{n^2})$ and has a solution $x^* = (1, 1, 0)^T$, such that $Ax^* = b$ or $\|Ax^* - b\|_2 = 0$. However, our problem formulation does not change.

In order to solve this problem using QAOA, we require 3 qubits. Using the formulation process detailed in the Appendix A and Eqn(12), the QUBO formulation we get is

$$F(x) = -12x_1 - 12x_2 - 13x_3 + 6x_1x_2 + 12x_1x_3 + 12x_2x_3. \tag{B.3}$$

The constant value that didn't make it to the QUBO here is $\|b\|_2^2 = 18$. Converting the QUBO into Ising using Eqn(16), we get

$$F(\sigma) = -1.5\sigma_1 - 1.5\sigma_2 - 0.5\sigma_3 + 1.5\sigma_1\sigma_2 + 3\sigma_1\sigma_3 + 3\sigma_2\sigma_3. \tag{B.4}$$

The offset when going from QUBO to Ising is -11. Therefore, the Ising ground state energy for this problem is $-18 - (-11) = -7$. Now let us assume that we are designing a circuit for QAOA where $p = 1$. So for a given pair of angles $(\beta, \gamma)$ a circuit would look like the one shown down below in Eqn(B.5).

The results of this circuit will be used to calculate the expectation value $C(\gamma, \beta)$. Based on the expectation value, a new pair of $\beta$ and $\gamma$ will be calculated using a classical black box optimization algorithm (like ImFil). These new angles will be fed into another such circuit till the optimization loop converges.

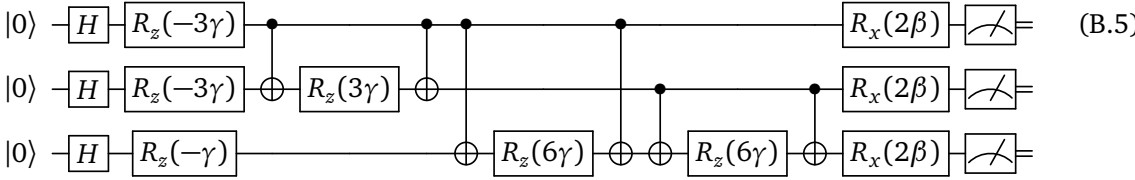

$$\tag{B.5}$$

The results from Eqn (B.5) would be classical bitstrings when measured in the standard basis. In order to calculate the energy or cost of a particular bitstring with respect to the Ising Cost function Eqn(3), we would first need to substitute a 1 for each 0 and -1 for each 1 in the bitstring. In short, this is because $\hat{\sigma}^{(z)}$ describes a quantum state to have an energy of $+1$ for $|0\rangle$ and $-1$ for $|1\rangle$ (in arbitrary units). For our example, if we measure a bitstring $\xi$ to be $\{\xi_1 = 0, \xi_2 = 0, \xi_3 = 1\}$ the equivalent Ising set would be $\{\sigma_1 = 1, \sigma_2 = 1, \sigma_3 = -1\}$. Using Eqn(B.4), we get back an energy of -7. This way, we can calculate $C(\gamma, \beta)$ for diagonal $\hat{C}$ Hamiltonians by averaging the energies of all measured bitstrings.

## C  Implementing two-qubit interactions on a QPU

As mentioned in Section 3.2.1, most practical quantum computers would not have all to all qubit connectivity. But if the problem that we need to solve on a quantum device requires dense connectivity (such as the BLLS), we need SWAP gates for allowing distant qubits to interact with one another. Let us first describe the SWAP gate in its matrix form

$$\text{SWAP} = \begin{pmatrix} 1 & 0 & 0 & 0 \\ 0 & 0 & 1 & 0 \\ 0 & 1 & 0 & 0 \\ 0 & 0 & 0 & 1 \end{pmatrix}. \tag{C.1}$$

Diagrammatically, we can decompose it with CNOT gates as

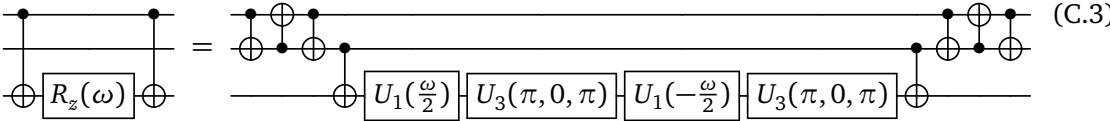

$$\text{SWAP} \qquad\qquad\qquad\qquad\qquad\qquad\qquad (C.2)$$

To illustrate how this takes place, consider a hypothetical device where every qubit is only connected to the adjacent qubit in a line. Thus in order to realize the gates described on the LHS of Eqn(C.3), one way is to SWAP between the top and the middle qubit so that it could interact with the bottom qubit.

$$(C.3)$$

After our desired two-qubit interaction takes place, the top and middle qubits are swapped again, returning the logical qubits to their original place. Of course, there are other methods (involving SWAP) that the compiler may take to realize the original unitary operations to be performed. In Eqn(C.3), we also show the decomposition of the $R_z$ gate as defined in Eqn(28).

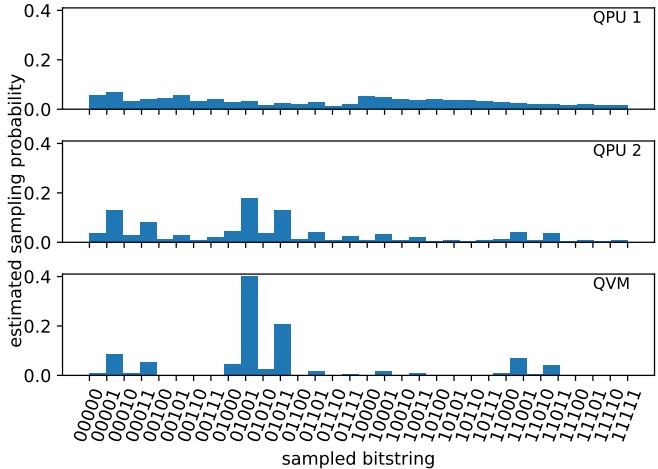

Figure 8: A typical instance of QAOA-circuit optimized probability distributions of the measured qubit bitstring output. We compare the result for IBM Q devices: `ibmq_london` (TOP) and `ibmq_athens` (MID) versus the qiskit shot-based simulator (BOTTOM), for the same problem instance for a total of 10240 shots. This figure represents data from an experiment done in Section 3.3.3.

# D  Probability distribution of running a QAOA circuit for BLLS on a real QPU

Figure 8 shows an example of the distribution we get from running a QAOA circuit with optimized (and fixed) $\beta$ and $\gamma$ angles. We run the circuit for a total of 10240 shots on a noiseless QVM (1024 shots over 10 queries to the QPU), the `ibmq_london` and `ibmq_athens` QPUs.

The simulation suggests a few bitstrings with a high probability of being measured (with the ground state having the highest) With `ibmq_athens` we see the QPU noise damping the highest probability strings, whereas the distribution from the `ibmq_london` quantum processor being more evenly spread out, with the ground state not having a significantly high probability of being measured. This can be attributed partially to `ibmq_athens` having a quantum volume (QV) [91] of 32 as opposed to `ibmq_london`'s QV of 16. Other factors that may have made a difference are the coupling map of the devices, and a shorter gate depth for the former.

# E  Curve fit and simulated annealing details

## E.1  Curve fit

In this part, we describe the details about the curve fits that appear in Figure 2 and 4. We found that for Figure 2, the polynomial curve of type $a \times n^b$ for $b = 0.85$ fits the data moderately well, Table 1 shows further information.

We can see from Table 1, that the fit error (a relative error value) doubles when we go from $p = 2$ to $p = 3$. However, when we explored other values for the exponent, we discovered that $b \approx 1.26$ reduces the fit error of QAOA depth $p = 3$ the most to about 0.13, but increases the error fit of $p = 1$ and 2 to about 0.22 and 0.17 respectively. On the other end, a value of $b \approx 0.65$ reduces the fit error for $p = 1$ the most to about 0.04 but raises $p = 2$ and $p = 3$ to around 0.12 and 0.26 respectively. Thus the $b = 0.85$ value is a compromise that works the best for all QAOA depths we consider in this paper. Further research on more problem sizes $n$ may help to fit better growth curves and get a more definitive value for the exponent $b$.

Table 1: Curve fit details for various QAOA depths from Figure 2 where the curve modeled is of the type: $a \times n^b$ for $b = 0.85$. All numerical values are rounded to a precision of 4 decimal places.

| QAOA Depth | Coeff. a | Fit Error |
|:---:|:---:|:---:|
| $p = 1$ | 0.0419 | 0.0852 |
| $p = 2$ | 0.0226 | 0.0911 |
| $p = 3$ | 0.0187 | 0.1911 |

Table 2: Curve fit details for various QAOA depths from Figure 4 where the curve modeled is of the type: $1 - (1 - a/2^{bn})^k$ at $k = 1$ for QAOA and $k = 10$ for simulated annealing (and extrapolated to $k = 10$ and $k = 1$ respectively for plotting purposes). All numerical values are rounded to a precision of 4 decimal places.

| Optimization Method | Coeff. a | Coeff. b | Fit Error |
|:---:|:---:|:---:|:---:|
| QAOA, $p = 1$ | 1.5968 | 0.3967 | 0.0655 |
| QAOA, $p = 2$ | 1.7127 | 0.3090 | 0.0247 |
| QAOA, $p = 3$ | 1.8926 | 0.3013 | 0.0380 |
| Sim. Anneal (1st attempt) | 0.6359 | 0.1397 | 0.0217 |
| Sim. Anneal (selected for plot) | 1 | 0.1786 | 0.0312 |

The per query and cumulative success probability curves for QAOA and simulated annealing as seen in Figure 4 fits the curve type $1 - (1 - a/2^{bn})^k$ well (which turns out to be just $a/2^{bn}$ for $k = 1$). For simulated annealing, we fix $a = 1$ such that $\mathcal{P}_{\text{eff}} \to 1$ as $n \to 0$, this is because $a \approx 0.63$ when its value is calculated in the curve fitting process, which prevents $\mathcal{P}_{\text{eff}}$

being 1 at $n = 0$. The increase in error when $a = 1$ is negligible for the simulated annealing curve. However, it is not so negligible in the case of QAOA, and therefore we don't fix the value of $a$ to 1. This results in the per-query curves to go above 1 and cumulative curves to be less than 1 (after reaching 1) for $n < 3$. We consider this to be a modeling artifact and fix the success probability to 1 for $n < 3$ for both cases.

### E.2 Simulated annealing experiments

The simulated annealing results referenced in Section 3.4.4, Figure 4 and Table 2 were conducted for BLLS problems of sizes 3 to 20. An initial and final temperature of $T_0 = 100$ and $T_f = 0.01$ (in arbitrary units) were chosen respectively. We define and use a exponential decay schedule with temperature at iteration $i$ as

$$T_i = T_0 \Big( \frac{T_f}{T_0} \Big)^{i/k}. \tag{E.1}$$

Each problem was run 1000 times with a set of randomization seeds for initialization of the Monte Carlo process. Each simulated annealing run was done with $k = 10$ steps. Our (cumulative) success probability is the count of all the runs where we end up with the ground state divided by 1000.

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
