# Peer review of "Quantum Approximate Optimization for Hard Problems in Linear Algebra"

_SciPost Physics Core, doi:SciPost Phys. Core 4, 031 (2021)_

## Round 1 · Referee Report · Anonymous (Referee 1) · 2021-7-27

Strengths

  1. Good general introduction to the problem
  2. Detailed discussion of the simulations
  3. Suggestions for further investigations

Weaknesses

  1. It would be useful if the authors present some qualitative discussion (or references) for the expected scaling of the results with the depth/size of the quantum circuits vs observed scaling.
  2. The paper may benefit from having a clear outline of the results in the introduction.
  3. There are a few points which should be clarified (see below)

Report

The authors apply the theory developed by Farhi et.al. [4] to the studies of binary linear least squares problem using IBM Q quantum computers (with 5 qubits) and compare performance (and error) of these calculations vs. numerical simulations on a classical computer with and without noise. They find that their quantum algorithm seem to perform significantly worse than Simulated Annealing for quantum circuits with the depths p<=3.

This paper contributes to ongoing efforts in developing and testing quantum algorithms for NISQ quantum computers, and therefore I believe it would be useful for the growing community of people working in this area.

However, I am not convinced that the current version of the manuscript satisfies high acceptance criteria of SciPost Physics Core Journal.

There is an interesting result reported in the paper, namely that Simulated Annealing outperforms the algorithm proposed by Farhi et.al., and I would suggest that the authors explore this result in more detail with e.g. higher circuit depth simulations on a classical computer, and if possible supplement these numerical calculations with some qualitative discussions.

Requested changes

  1. Add more results and discussions on the question of Simulated Annealing vs QAOA algorithm. Specifically one can compare the calculations of the Simulated Annealing vs QAOA algorithm for much larger circuit depths and bigger system sizes (with simulations done on a classical computer).

  2. Perhaps reorganise the paper slightly for better readability. Present an outline of the main results in the beginning of the paper.

Some other points:

  1. In the sentence "Because it has been shown to be NP- Complete in nature [41]" add references to original theoretical papers ([41] is a popular review), e.g. F Barahona 1982 J. Phys. A: Math. Gen. 15 3241.

  2. On the point of terminology. The Hamiltonian in Eq. 5,6 is not a quantum Ising, but a classical Ising Hamiltonian. The authors also use the term "Quantum Volume", and it would be useful to add the definition of this term in the text for general physics audience.

  3. In Fig. 1 the "qvm", and "qpu" are not defined.

  4. In Fig. 1 it is not clear why the numerical simulation ("statevector") does not converge to the exact value.

  5. The authors should explain what do they mean exactly by simulations with "the statevector backend" and explain the details of the algorithm for the noisy simulation on a classical computer.

  6. In Fig. 2, is there some qualitative understanding of the scaling of the error with the depth of the circuit?

  7. I am not sure to understand this statement "At the time of writing, the IBM Q devices we tested on did not approximate the theoretically-optimal QAOA result distribution very well, but it still finds the best solution every time." What do the authors mean when referring to the "best solution", see e.g. Fig. 1 where the "qpu" results seem to show a very poor convergence.

  8. In Fig. 7 it is not clear to me why "qvm exact energy" has a relative error of 0.2.

  9. What do the authors mean by this "we extract the noise model of a qpu (ibmq athens in our case )".

  • validity: ok
  • significance: ok
  • originality: ok
  • clarity: ok
  • formatting: good
  • grammar: good

Author:  Ajinkya Borle  on 2021-09-04  [id 1735]

(in reply to Report 1 on 2021-07-27)
Category:
remark
answer to question

We thank the reviewer for taking the time to thoroughly review our manuscript. We believe that the assessment has been done very fairly . We want to clarify on some of the weaknesses and requested changes

(In weaknesses)
“It would be useful if the authors present some qualitative discussion (or references) for the expected scaling of the results with the depth/size of the quantum circuits vs observed scaling”

To the best of our knowledge, there are no rigorous derivations for this problem type but more importantly, the same holds true for almost all problems where the (heuristic in nature) QAOA algorithm is applied. With our work, we are the first ones to model Ising based linear algebra problems onto gate model machines (to be solved with QAOA). Similar works exist on Quantum Annealing, and an important factor for performance (apart from noise) is the minimum gap between the ground state and the first excited state of the Hamiltonian, and we mention those in our manuscript.

(In requested changes)
“1. Add more results and discussions on the question of Simulated Annealing vs QAOA algorithm. Specifically one can compare the calculations of the Simulated Annealing vs QAOA algorithm for much larger circuit depths and bigger system sizes (with simulations done on a classical computer)”

We understand the concern behind this requested change. However, simulations of p > 3 and larger sizes with our methodology would become computationally infeasible as the current research is done on a limited computational budget (laptops for simulations instead of HPC clusters). Our work is one of the firsts in the intersection of hard problems in linear algebra as applied to QAOA and we believe that future research would greatly benefit from our work being published (leading to subsequent funded works being done with greater circuit depths and larger systems).

However, we will comment on the possibility of when we may expect an advantage based on other related works done on QAOA. Such as this one which suggests it would require the problem size to be in the 100s for QAOA to have an advantage (based on MAXCUT) : Guerreschi, Gian Giacomo, and Anne Y. Matsuura. "QAOA for Max-Cut requires hundreds of qubits for quantum speed-up." Scientific reports 9.1 (2019): 1-7.

“2. Perhaps reorganise the paper slightly for better readability. Present an outline of the main results in the beginning of the paper.”

We thank the reviewer for this suggestion, we’ll make the requested changes as best applicable.

(Other points)
“1. In the sentence "Because it has been shown to be NP- Complete in nature [41]" add references to original theoretical papers ([41] is a popular review), e.g. F Barahona 1982 J. Phys. A: Math. Gen. 15 3241.”

We appreciate the reviewer’s expertise in this field (and this suggestion). This was very useful.

Points 2 and 3.

We will make the required clarifications.

“4. In Fig. 1 it is not clear why the numerical simulation ("statevector") does not converge to the exact value”

The numerical simulation with “statevector” doesn’t converge to the exact ground state which is not completely surprising because this case was done with a QAOA depth of p=1. In other words, it is the best that QAOA can do with the minimum required circuit depth. By Eqn (45) of Farhi’s original paper (arXiv:1411.4028) we only reach the exact value as p approaches infinity (in the general case). We’ll clarify it in the text the best we can.

“5.The authors should explain what do they mean exactly by simulations with "the statevector backend" and explain the details of the algorithm for the noisy simulation on a classical computer”

We apologize for the confusion. “The statevector backend” is essentially the name for the noiseless waveform simulation with respect to the QISKIT SDK. We’ll clarify this and provide details where needed.

“6. In Fig. 2, is there some qualitative understanding of the scaling of the error with the depth of the circuit?”

The relative error is of the expectation value with respect to the ground state energy. And thus, we attribute its growth in polynomial and not exponential terms as there being spin configurations that have energies that are close to the ground state, or at the very least, not extremely dissimilar. We mention this in 3.4.2, but we’ll see if this can be revised.

“7. I am not sure to understand this statement "At the time of writing, the IBM Q devices we tested on did not approximate the theoretically-optimal QAOA result distribution very well, but it still finds the best solution every time." What do the authors mean when referring to the "best solution", see e.g. Fig. 1 where the "qpu" results seem to show a very poor convergence”

In each iteration of the optimization, the device is run for the same QAOA parameters for a number of times (shots) to approximate the expectation value for those parameters. Currently, the IBM Q devices are not able to produce the actual distribution for the parameters too well (see Fig 8 for reference), and this affects the performance of the classical black box optimizer being used (IMFIL).

However, in our tests, through random probability (or call it luck), it stumbled upon the qubit configuration that corresponds to the ground state of the problem Hamiltonian (which is the classical Ising Hamiltonian we transform our domain problem into). This is what we meant by finding the best solution though in reality the convergence is very poor. We’ll clarify where needed in the manuscript.

“8. In Fig. 7 it is not clear to me why "qvm exact energy" has a relative error of 0.2”

Here qvm exact energy refers to the energy from the noiseless waveform simulation, and the reason that it has a relative error of 0.2 is because its done with a QAOA depth p=1. This is a similar situation as with the concern with Fig 1. We’ll make an attempt in the manuscript to articulate this better.

“9. What do the authors mean by this ‘we extract the noise model of a qpu (ibmq athens in our case )’”

The IBM Q API allows us to generate a noise model based on its QPU. As our template, we chose the noise model of ibmq athens as a starting point. However, we can see that the statements in the manuscript can confuse people and need to be explained a bit better. We thank the reviewer for pointing this out.

Once again, the authors thank the reviewer for taking the time to go through the manuscript.

Sincerely
The Authors

---

## Round 1 · Referee Report · Anonymous (Referee 2) · 2021-8-18

Strengths

1- Detailed exploration of applicability of Quantum Approximate Optimization Algorithm to Binary Linear Least Squares on Noisy Intermediate Scale Quantum computers. 2- Pedagogical text with good review of background.

Weaknesses

1- Limitation to (very) small problem sizes. 2- Tendency to be simplistic and verbose. 3- Formatting and typography could be improved.

Report

The authors explore the Quantum Approximate Optimization Algorithm for Binary Linear Least Squares. This algorithm may form the building block for several other hard problems in linear algebra, such as the Non-negative [Binary] Matrix Factorization. Experiments were done on noiseless quantum simulators, a simulator including a device-realistic noise-model, and two IBM 5-qubit machines. This work is more of an algorithmic nature, but the hardware aspect may justify publication in a physics journal.

I am generally interested in the field, but not an expert. As such, I cannot help the feeling that the problems under investigation are so small that a classical computer would easily compete with or even outperform a quantum computer, in particular a noisy one. For example, I believe that for $n=10$, one could rapidly enumerate all $2^{10}=1024$ possible binary solution vectors $x$ in Eq. (1) and find the optimal solution (not to mention the $32$ cases allowed for $n=5$, i.e., the case explored on the two IBM machines). Even a simulated-annealing solution of the problem would probably outperform such a direct search only for much larger $n$, i.e., when a direct search is no longer possible. The idea of O’Malley, Vesselinov et al. [17] to use (run?)time for benchmarking sounds interesting from this point of view. Note that if the message is that presently available "quantum computers" fail to perform properly even at very small $n$, this is fine with me. Nevertheless, an estimate of the order of magnitude of $n$ where quantum computers might outperform classical ones would be helpful to put the work into context.

Switching to presentation issues, the manuscript is a bit verbose. For an interested outsider, a detailed and pedagogical description may indeed helpful. However, Appendix A may be going a bit too far. After all, Eqs. (35) and (36) duplicate Eqs. (13) and (14), and Eqs. (31)-(34) are an undergraduate physics exercise (note that readers of SciPost Physics may be expected to have a physics background). Likewise, given that Eq. (17) is represented in terms of the diagonal matrices (7), commutativity of the components is obvious and does not need to be justified by Ref. [4].

There are a number of further minor items that I list among "Requested changes". Once these changes have been made, I expect the manuscript to be suitable for publication in SciPost Physics Core.

Requested changes

1- Add a comment on the order of problem size $n$ for which a quantum advantage might indeed be expected. 2- The authors present two Algorithms. I believe that they use pseudo-code, but an explicit statement to this effect would be helpful. 3- Remove reference to [4] in line below Eq. (17). 4- The manuscript has obviously been reformatted from a different style file. However, necessary replacements of "top" and "bottom" by "left" and "right" for the panels of Figs. 3, 4, and 6 still need to be made. Likewise, Eq. (44) is too wide. 5- Remove Appendix A. 6- Line 2 of appendix D: is there a spurious "0" at the end of "10240"? 7- The manuscript would generally benefit from careful proofreading, in particular: a) Spurious spaces before punctuation marks. Sometimes, spaces are also missing (e.g., after "[90]." on page 19), but this is less frequent. b) Spelling errors (e.g., "ExperimemtAL methods" as heading of section 3, spurious hyphen in "no-noise" in heading of section 3.4.2?) or incorrect words (e.g., "factors" instead of "function" in the caption of Fig. 6?). c) Make sure that all acronyms are properly defined (e.g., "PCA" - maybe this acronym is actually not needed since it occurs only once). d) Spurious articles such as "a the" on page 4. e) "Where" after Eq. (4) just after "where" in the equation. f) In the paragraph below Fig. 5, I believe that "(for" contains an unclosed parenthesis. 8- If a field has a strong preprint culture, a large fraction of preprint references is normal, but some preprints among the references have been published, in particular: [19] in PLOS ONE 2021 16(1): e0244026. [32] in SIAG/OPT Views and News 25 (1), pp. 7-16 (2017) [70] in Science Bulletin, https://doi.org/10.1016/j.scib.2021.06.023 [72] in Quantum 5, 483 (2021) [77] in Phys. Rev. Lett. 126, 070505 (2021) [83] in Phys. Rev. A 103, 042612 (2021) [88] in Nature Physics 17, 332-336 (2021). Note that the list of authors of this reference starts with "Harrigan" and not "Arute". 9- If conference contributions are the only source, this cannot be helped, but at least in the case of Ref. [76], the article Phys. Rev. X 10, 021067 (2020) might be preferable over a conference abstract. 10- Titles in the references are useful, but in the present case, they are subject to excessive lowercasing (names such as "Ising" start with a capital letter, acronyms should not be systematically lower-cased, etc). Journal names sometimes also contain spurious lowercasing; in Ref. [31] even a name of an author ("Brien") is lower-cased. By contrast, in Ref. [65] there is excessive uppercasing. Finally, the title of Ref. [83] contains a strange "p¿1". Probably most of that could be fixed during production with the aid DOIs (i.e., please add), but given the length of this list, I think that the authors should make an effort to improve on these items.

  • validity: good
  • significance: good
  • originality: good
  • clarity: good
  • formatting: good
  • grammar: good

Author:  Ajinkya Borle  on 2021-09-04  [id 1736]

(in reply to Report 2 on 2021-08-18)
Category:
remark
answer to question

We thank the reviewer for their timely and thorough review of our manuscript. In particular, we appreciate them catching some of the more glaring typographical errors in the manuscript. Based on the report, we want to comment on certain aspects for clarification.

With regards to the small size of the problems and the seeming trivialness of them. When it comes to the simulations, we were limited by our computational budget to problems of size n<= 10. But budget constraints aside, we believe that even by considering problems of sizes this small, we are able to do justice in the first ever attempt of studying hard linear algebra problems using QAOA (via the classical Ising model). The reason this has considerable relevance in the field is because using Ising based formulations on a Quantum Annealer to solve linear algebra problems has a good deal of popularity amongst researchers exploring emerging computational techniques .

Response to other requested changes :
“1. Add a comment on the order of problem size n for which a quantum advantage might indeed be expected.”

This is indeed going to be useful for the reader, we’ll comment and add the reference by the people over at Intel who did an analysis based on problem size n for MaxCut : Guerreschi, Gian Giacomo, and Anne Y. Matsuura. "QAOA for Max-Cut requires hundreds of qubits for quantum speed-up." Scientific reports 9.1 (2019): 1-7.

“2. The authors present two Algorithms. I believe that they use pseudo-code, but an explicit statement to this effect would be helpful”

We apologize for the confusion, we will make it explicit that it is pseudocode

“3. Remove reference to [4] in line below Eq. (17)”

Will do.

“4. The manuscript has obviously been reformatted from a different style file. However, necessary replacements of "top" and "bottom" by "left" and "right" for the panels of Figs. 3, 4, and 6 still need to be made. Likewise, Eq. (44) is too wide”

We thank the reviewer for catching such glaring errors, we’ll fix them all.

“5. Remove Appendix A”

We understand the concern with what may seem like verbosity. Because of quantum computing becoming a rapidly interdisciplinary field, we felt that certain concepts that might be obvious for physicists would perhaps be required to be mentioned for completeness. Thus although bordering on redundant, we feel that Appendix A is an important part of the manuscript.

However, we will trim it down and remove Eqn(35) and Eqn(36) as they just repeat Eqn(13) and Eqn(14).

“6. Line 2 of appendix D: is there a spurious "0" at the end of "10240"?”

No there isn’t, we wanted a large amount of shots for the machines to approximate the probability distributions as best as they can. But we’ll clarify it the best we can.

“7. The manuscript would generally benefit from careful proofreading…” along with 8 and 9.

We apologize for all the errors having made their way into the manuscript. We’ll be going through it again to rigorously remove them. However, in figure 6, we do mean factors, but it is also the case where we need to reword the sentence for it to make sense.
With regards to references, we thank the reviewer for bringing several key errors to our attention, which we will fix. Going forward, it is clear that we need to carefully go over the references as they appear in the manuscript as well. This field does have a heavy arxiv culture . However, we do make a conscious effort to try and cite the peer-reviewed version of a reference rather than the arxiv version if available.

Once again, the authors thank the reviewer for taking the time to go through the manuscript.

Sincerely
The Authors

---

## Round 2 · Referee Report · Anonymous (Referee 1) · 2021-9-5

Report

I find the author's responses to my questions/comments satisfactory and recommend the paper for publication in SciPost.
  • validity: -
  • significance: -
  • originality: -
  • clarity: -
  • formatting: -
  • grammar: -

Author:  Ajinkya Borle  on 2021-10-07  [id 1821]

(in reply to Report 1 on 2021-09-05)

We thank the referee for their decision (and their time and effort).

---

## Round 2 · Referee Report · Anonymous (Referee 2) · 2021-10-4

Strengths

1- Detailed exploration of applicability of Quantum Approximate Optimization Algorithm to Binary Linear Least Squares on Noisy Intermediate Scale Quantum Computers. 2- Test of real cloud-accessible quantum-computing devices. 3- Pedagogical text with good review of background.

Weaknesses

1- Still on the verbose side. 2- Still a number of typographic errors.

Report

With their revised version, the authors have addressed a big fraction of the previous criticism. I would still have number of minor, in particular typographic complaints (see "Requested changes"), but I think that these could be implemented during the production, respectively proof stage. Thus, I recommend the present version for publication in SciPost Physics Core.

Requested changes

1- Still a relatively high number of spacing errors (such as spaces before a colon ":" that should not be there). 2- The acronym "HHL" in the second line of section 2.2 can be identified from the list of authors of Ref. [53], but should really be explained in the text (or avoided, given that it only occurs once). 3- In relation to the "other point" 2 of the other Report (Report 1), there is still a "quantum Ising" in the sentence preceding the classical Ising model of Eq. (17). 4- Better insert "energy" after "ground state" in the line below Eq. (38)? 5- The references still have no DOIs. This has not only been previously requested by me, but is a highlighted item in the author guidelines, see https://scipost.org/SciPostPhys/authoring. 6- Upper- and lower-casing of titles has not really been improved. This could probably be fixed with the aid of DOIs, but see preceding item. 7- Ref. [60] has in the meantime appeared in Phys. Rev. A 104, 032422 (2021). 8- The full journal reference for Ref. [74] is Science Bulletin 66, 2181 (2021), compare again item 5. 9- Ref. [86] has a DOI (!), but the list of authors is very different from what I find on zenodo. 10- Ref. [87] still has the strange "p¿1" in its title that I already pointed out in my first report.

  • validity: good
  • significance: good
  • originality: good
  • clarity: good
  • formatting: good
  • grammar: good

Author:  Ajinkya Borle  on 2021-10-07  [id 1822]

(in reply to Report 2 on 2021-10-04)

We thank the referee for their recommendation. We apologize for the typographical errors, especially the ones pointed out before. We'll take this learning to do better henceforth in general.

---

## Round 2 · Author Response

We thank the editor and the referees for taking the time to review this work. We did our best to incorporate all the changes from the referees (and have replied to each of the reports).

---

## Round 2 · List of Changes

Following is a list of changes now in the manuscript.

1.Commented on the expected scaling of the problem based on other works done in the field with the QAOA algorithm.
2.Added an outline of results in the introduction section.
3.Fixed the references, capitalization and spelling errors based on report 2.
4.Defined several terms that were previously taken as implicit.
5.Trimmed duplicate equations.
6.Addressed points of confusion expressed by both the reports.

---

## Editorial Decision

published